# Multisystem Inflammatory Syndrome in Adults Associated with Recent Infection with COVID-19

**DOI:** 10.3390/diagnostics13050983

**Published:** 2023-03-04

**Authors:** Ondrej Zahornacky, Štefan Porubčin, Alena Rovnakova, Pavol Jarcuska

**Affiliations:** Department of Infectology and Travel Medicine, Faculty of Medicine, Louis Pasteur University Hospital, Pavol Jozef Šafarik University, 04190 Košice, Slovakia

**Keywords:** COVID-19, infection, multisystem, inflammatory, MIS-A

## Abstract

Multisystem inflammatory syndrome in adults (MIS-A) is an uncommon but severe and still understudied post-infectious complication of COVID-19. Clinically, the disease manifests itself most often 2–6 weeks after overcoming the infection. Young and middle-aged patients are especially affected. The clinical picture of the disease is very diverse. The dominant symptoms are mainly fever and myalgia, usually accompanied by various, especially extrapulmonary, manifestations. Cardiac damage (often in the form of cardiogenic shock) and significantly increased inflammatory parameters are often associated with MIS-A, while respiratory symptoms, including hypoxia, are less frequent. Due to the seriousness of the disease and the possibility of rapid progression, the basis of a successful treatment of the patient is early diagnosis, based mainly on anamnesis (overcoming the disease of COVID-19 in the recent past) and clinical symptoms, which often imitate other severe conditions such as, e.g., sepsis, septic shock, or toxic shock syndrome. Because of the danger of missing the treatment, it is necessary to initiate it immediately after the suspicion of MIS-A is expressed, without waiting for the results of microbiological and serological examinations. The cornerstone of pharmacological therapy is the administration of corticosteroids and intravenous immunoglobulins, to which the majority of patients clinically react. In this article, the authors describe the case report of a 21-year-old patient admitted to the Clinic of Infectology and Travel Medicine for febrility up to 40.5 °C, myalgia, arthralgia, headache, vomiting, and diarrhea three weeks after overcoming COVID-19. However, as part of the routine differential diagnosis of fevers (imaging and laboratory examinations), their cause was not clarified. Due to the overall worsening of the condition, the patient was transferred to the ICU with suspicion of developing MIS-A (he met all clinical and laboratory criteria). Given the above, reserve antibiotics, intravenous corticosteroids, and immunoglobulins were added to the treatment due to the risk of missing them, with a good clinical and laboratory effect. After stabilizing the condition and adjusting the laboratory parameters, the patient was transferred to a standard bed and sent home.

## 1. Introduction

Multisystem inflammatory syndrome was first described as a nosological entity in 2020, initially mainly in a group of pediatric patients (as MIS-C). Later, the first cases of this disease also began to appear in a group of adult patients (MIS-A). In adults, the clinical course is extremely variable, with primarily febrile, systemic inflammation, often with signs of shock and organ involvement. Therefore, the differentiation from Kawasaki disease is necessary as part of the differential diagnosis, especially in pediatric patients [1].

The pathophysiology of the disease still needs to be precisely discovered. The SARS-CoV-2 coronavirus induced a dysregulated pathological immune response in the host, resulting in systemic vasculitis and multiple acute organ damage [2].

Complement activation with subsequent capillary deposition of immunocomplexes also comes into consideration [3].

Diagnosing the disease is quite challenging due to its varied clinical symptomatology. According to the Centers for Disease Control (CDC), several basic criteria must be met to be diagnosed with MIS-A. MIS-A is defined as a severe illness requiring hospitalization for more than 24 h in persons aged 21 years or older or an illness ending in death based on clinical and laboratory signs. The most important clinical symptom is a fever above 38 °C (subjective or documented fever) for ≥24 h prior to a hospitalization or within the first three days of hospitalization; moreover, at least three of the following clinical criteria must occur prior to hospitalization or within the first three days of hospitalization. At least one must be a primary clinical criterion (Table 1) [4].

Primary clinical criteria

Serious involvement of the heart: myo-, pericarditis, dilatation or aneurysm of the coronary arteries, or new dysfunction of the right or left ventricle, atrioventricular block II.-III. degree or ventricular tachycardia;rash and nonpurulent conjunctivitis.

Secondary clinical criteria

Newly developed neurological signs and symptoms: encephalopathy in a patient without previous cognitive deficit, convulsions, meningeal symptoms, peripheral neuropathy;Shock or hypotension not caused by medication (sedation);Abdominal pain, vomiting, diarrhea;Thrombocytopenia.

Laboratory evidence: evidence of SARS-CoV-2 infection, elevation of inflammatory markers

Elevated value of at least two of the following: C-reactive protein (CRP), ferritin, IL-6, erythrocyte sedimentation rate, procalcitonin (PCT);Positive test for SARS-CoV-2 during illness using RT-PCR, serology, or antigen detection.

## 2. Case Report

A 22-year-old patient, who was not treated for anything prior to testing, was examined at the outpatient department of the infectious disease clinic for fever lasting 3 days with a maximum temperature of up to 40.2 °C, as well as myalgia, arthralgia, headache, a dry cough with dyspnea, and vomiting. The patient reported neutrophilia, lymfophenia, or thrombocytopenia. Neutrophils: 93.2%; Lymphocyte 3.30%; Platelets 51 × 10^9^/L He describes the clinical course of the COVID-19 disease as a mild–subfebrile dry cough. The patient was vaccinated once against COVID-19.

The patient had a meningeal skin free of pathological efflorescences, was seized, objectively febrile (40.0 °C), cardiopulmonary compensated (104/70 mmHg), tachycardic (regular heartbeat, frequency 129/min.), without lymphadenopathy, and was otherwise unremarkable at the initial examination (SOFA score 2). The results of the lungs’ ultrasonography (USG) test were negative for pathological abnormalities, and there were no further anamnesis or epidemiological results (unprotected sexual contact, including sex with men, contact with an infectious disease, IV drug addiction, tattooing, piercing, stay in nature, source of water and food, a bite of a tick, etc.). Due to the general condition, the patient was admitted to the infectious disease clinic with a fever of unknown origin. The nasopharynx was tested using PCR for COVID-19 at the entry, and the results showed a positive result and a ct cycle of 33.54, which was considered a requirement for recovering from PCR influenza A/B negative.

On admission, an electrocardiogram (ECG) assessment was recorded, which described sinus tachycardia without ischemic or other pathological changes. The results of the initial laboratory tests first indicate a viral aetiology of the disease; therefore, complex symptomatic treatment was started.

As part of the differential diagnosis of the febrile state, a host of serological examinations were carried out, and blood cultures were taken repeatedly. Regarding influenza PCR A/B negative, the supplemented ultrasound examination of the abdomen describes only mild splenomegaly. After two days of hospitalization, fever persisted, thrombocytopenia worsened, and IL-6 and CRP increased (Table 1). Diarrhea and abdominal pain appeared in the clinical picture. A microbiological examination of the stool did not detect an infectious agent (cultivation, detection of antigen of viruses and *Clostridioides difficile* toxin). For leukopenia in the differential blood count (3.30 × 10^9^/L), individual subpopulations of lymphocytes were also examined. The number of CD4 + T lymphocytes was critically low (CD4 + 0.10 × 10^9^/L, CD3 + 0.14 × 10^9^/L, CD8 + 0.04 × 10^9^/L, natural killers 0.02 × 10^9^/L), which is why we also considered human immunodeficiency virus (HIV) infection as part of the differential diagnosis. Due to the danger of not receiving therapy, a combination anti-infective regimen that includes the prophylaxis of opportunistic infections (cefotaxime, metronidazole, co-trimoxazole, fluconazole, and azithromycin) is recommended.

Over the next 2 days, fevers up to 40 °C persisted, and dyspnea with hyposaturation (sp O2 92%), hypotension, and tachycardia appeared. A maculopapular exanthema appeared on the chest, and eyelid edema and nonpurulent conjunctivitis appeared. Control ultrasonography of the lungs was performed, and it revealed confluent B-lines on both sides, sinus tachycardia and bigeminal ventricular extrasystoles on the ECG, worsening thrombocytopenia in the lab (Table 1, Figure 1), and an increase in inflammatory markers (CRP, IL-6, PCT), as well as troponin T and NTproBNP (Table 1).

Antibodies against HIV are repeatedly negative (ELISA test) (Table 2). An echocardiographic examination was performed acutely based on the suspicion of perimyocarditis, without pathological findings. Due to the deterioration of the clinical condition, the patient was transferred to the intensive care unit to monitor vital functions based on the suspicion of the development of MIS-A (SOFA score 5). After an overall evaluation of the condition, the patient met the criteria necessary for diagnosing MIS-A (primary and secondary). Intravenous immunoglobulins in a cumulative dose of 2 g/kg (130 g in total) and methylprednisolone in a dose of 1 mg/kg (5 days in total) were added to the treatment; due to the impossibility of excluding another etiology of the disease, especially septic shock, the antibiotic treatment was changed to piperacillin/tazobactam with gentamicin.

A high-resolution computed tomography (HRCT) examination of the chest was also performed due to the deterioration of the findings on the USG of the lungs, as well as dyspnea and a decrease in saturation. This examination revealed dorsally situated pleural effusions reaching a width of 25 mm in the anteroposterior plane with a predominance on the right, along with discrete subsegmental dyslectases existing on the effusions, fluidopericard of up to 19 mm with and without morphological signs of pneumonia caused by bacteria, viruses, or pneumocystis, without interstitial lung edema, and without cardiac undercompensation in the event of a probable myopericarditis (Figure 2).

A hematologist was consulted to facilitate a decrease in the number of platelets, who ruled out hemolytic–uremic syndrome and thrombotic thrombocytopenic purpura. Immunomodulation treatment included consultation with an immunologist, who recommended the continuation of the applied treatment without change. In case of the progression of the condition, he recommended adding biological treatment, i.e., anakinra or infliximab.

The patient’s condition improved upon receiving the aforementioned treatment: the fever subsided, and laboratory parameters improved (Figure 1 and Figure 3). Considering the highly suspicious MIS-A, the favorable clinical and laboratory effect of the administered immunomodulating treatment, and after ruling out a focal bacterial infection (laboratory and imaging methods), we ended the antibiotic treatment.

The examination was completed by an internist, who identified a sinus rhythm without ventricular extrasystoles using an electrocardiogram (EKG). He recommended adding colchicine to the treatment due to the use of HRCT on the lungs, as well as fluidopericarde, and suspected myopericarditis (first day 1 mg, then 0.5 mg for a total of 2–3 months, depending on the condition), for which he recommended bisoprolol (5 mg daily); moreover, the magnetic resonance of the heart was supplemented to rule out myocarditis definitively. However, this examination method is not available in our hospital. Therefore, the patient was booked as an outpatient in another city for this examination. After stabilizing the patient’s general condition and adjusting laboratory parameters, the patient was discharged to outpatient care after 22 days of hospitalization.

He continued oral treatment with methylprednisolone with a gradual dose reduction, as well as with beta-blocker and colchicine. Moreover, he continued to be consulted by a cardiologist, immunologist, and internist. Ambulatory examination of the heart was conducted using magnetic resonance imaging. Morphologically, both ventricles and atria were observed without dilatation and hypertrophy, pericardium without thickening, and a discrete pericardial effusion of up to 5 mm was detected. In the T2 short-TI inversion recovery (STIR) sequence, in midventricular anteroseptal located in segment 8, there was a zone of slightly increased signal-residual inflammatory changes without an evident ventricular kinetics disorder. The appropriate values of the T1 times are shown in the T1 maps. In the T2 maps, in segment 8, T2 times were slightly increased to a maximum of 66 ms (the rest of the myocardium up to 50 ms), at rest perfusion without an observable perfusion disorder, after the administration of a contrast agent (gadolinium) without pathological enhancement in the myocardium, as well as without the presence of scar or fibrosis in the borderline systolic left ventricular function (ejection fraction 52%, end-diastolic volume 169 mL, end-systolic volume 81 mL) (Figure 4).

## 3. Discussion

Multisystem inflammatory syndrome represents a potentially life-threatening complication upon infection with COVID-19, the pathophysiology of which is not yet fully understood. The syndrome was first described in April 2020 in a group of children whose clinical symptoms resembled Kawasaki disease. Later, similar cases began to appear in adult patients, called MIS-A [5].

The interval between viral infection and the development of MIS-A varies in length (2–6 weeks), but it can be clinically manifested already during an acute SARS-CoV-2 infection. That is why it is unclear whether this is a manifestation of an acute infection or a post-infectious syndrome [6].

The clinical symptoms of the disease are diverse. Most of the patients described in the case reports had similar symptoms to the patient from our case report.

Patel et al. published a review of 221 patients with MIS-A worldwide. In this group of patients, the disease developed an average of four weeks after acute COVID-19 infection. The median age of patients in the monitored group was 21 years (19–34 years), 70% were men, and 58% had no other comorbidities. The main symptoms of MIS-A were fever (96%), hypotension (60%), cardiac dysfunction (54%), dyspnea (52%), and diarrhea (52%). Moreover, 57% of patients were admitted to the ICU, 47% required respiratory support, and 7% of patients with MIS-A died after hospital admission. Most patients with MIS-A (90%) had increased coagulopathy or inflammatory markers. The authors concluded that MIS-A with extrapulmonary multiorgan involvement was difficult to distinguish from both acute biphasic COVID-19 and the postacute sequelae of SARS-CoV-2 infection [7].

Behzadi et al. reported that men predominate in the case reports, the majority without any comorbidities. Fever and exanthema were among the most common clinical signs, while gastrointestinal clinical signs such as nausea, abdominal pain, and diarrhea were less frequently described [8].

In our case, high temperatures, headaches, myalgia, arthralgia, and a dry cough dominated. During hospitalization, nonpurulent conjunctivitis, maculopapular exanthema on the chest, and elevated laboratory markers of inflammation (CRP, PCT, IL-6) appeared. Diarrhea and abdominal pain in the patient described by us occurred only for 3 days, with spontaneous resolution after symptomatic treatment. The extrapulmonary manifestations of the disease progressed relatively quickly. Tachycardia with hypotension and dyspnea, in the laboratory, increased in TnT, and thrombocytopenia was added to the clinical picture. The clinical symptoms were reminiscent of septic shock, which necessitated the transfer of the patient to the intensive care unit. 

Mazumder et al. and Kobe et al., in their published case reports, described cases of MIS-A with a severe course that was complicated by the development of disseminated intravascular coagulopathy. In the case described by us, this complication did not develop during hospitalization (despite two positive criteria, namely, D-dimer and platelets, ISTH score: 3) [9,10,11].

Based on the clinical symptoms and laboratory results, the diagnostic reliability of our clinical case (Table 3) (according to the Brighton Collaboration definition for MIS-A and MIS-C) is level 1 (definitive case, Table 3) [12]. 

It is reported in the literature that MIS-A is often associated with cardiovascular damage, the manifestation of which is most often tachycardia, hypotension, and possibly a shocking state with a documented disorder of the ejection fraction of the left ventricle [13,14,15].

The same clinical signs of cardiovascular system involvement were also present in the case report described by us. However, the clinical condition did not progress to shock, and treatment with vasopressors was not necessary for an early diagnosis of the disease and the rapid initiation of adequate treatment.

Magnetic resonance of the heart is an imaging modality that is often mentioned in the literature and is mainly used for the diagnosis of myocarditis in MIS-A. The examination can confirm the presence of the diffuse inflammation of the myocardium and, at the same time, rule out another cause of its damage, such as ischemic or stress-induced cardiomyopathy [16].

In our case, the MRI examination of the heart was performed only one month after the patient was discharged to outpatient care (due to the unavailability of the examination in our city). Nevertheless, it was possible to capture the midventricular anteroseptal zone of a slightly increased signal as a sign of residual inflammatory changes in the myocardium and the borderline systolic function of the left ventricle. DeCuir et al. reported that inflammatory involvement of the myocardium and a reduction in the left ventricular ejection fraction occurs in up to 66% of patients with MIS-A [17].

Yao et al. reported that most patients with MIS-A had a negative PCR test for the presence of the SARS-CoV-2 virus at the initial examination. They emphasized that it is essential to enquire about recently overcome viruses as part of the differential diagnosis of hyperinflammatory conditions, given that a significant part of COVID-19 infections is either subclinical or asymptomatic. Equally beneficial is the serological examination of antibodies against the SARS-CoV-2 virus [18].

In our case, the patient had a positive PCR test for the presence of SARS-CoV-2 in a high ct cycle during the initial examination. We considered this result a condition after overcoming the disease in the recent past, which the patient also confirmed in their anamnestic response. Therefore, precisely because of PCR positivity has recently been shown as proof of overcoming the infection, we did not perform a serological test for anti-SARS-CoV-2 antibodies.

Several treatment options for MIS-A are described in the literature. According to the analysis results by Kunal et al., who investigated the treatment of MIS-A in a total of 79 patients, steroids (60.2%) and intravenous immunoglobulins (37.2%) were most often used in the treatment. Only about 10% of patients required biological treatment [19].

In our case, the patient was initially adequately resuscitated with fluids to maintain adequate perfusion of the tissues. Then, broad-spectrum antibiotics were administered due to the danger of missing the patient, since it was not possible to exclude with certainty another cause of the described condition, especially septic shock (even though it was very unlikely from a clinical point of view, as no focal infection was proven, and the results of the performed microbiological and imaging examinations were negative). Finally, when the condition worsened, and the MIS-A criteria were met, we followed the National Institute of Health’s recommendation for the treatment of MIS-C/MIS-A [20].

We added corticosteroids (methylprednisolone), intravenous immunoglobulins, and colchicine to the treatment, as well as a prophylactic dose of low-molecular-weight heparin (to prevent thromboembolic complications), along with broad-spectrum antibiotics and symptomatic treatment. According to the literature, preventing thromboembolism is essential, as most hyperinflammatory syndromes, including MIS-A, are associated with the development of these complications. Therefore, we administered a dose of low-molecular-weight heparin despite the decrease in the number of platelets, as there was no decrease below 50 × 10^9^/L [21]).

The patient’s condition improved clinically and in the laboratory after administering corticosteroids and intravenous immunoglobulins. Therefore, the administration of biological treatment was not necessary in our case.

## 4. Conclusions

Multisystem inflammatory syndrome in adults represents a severe complication of COVID-19, whose pathophysiology has not yet been clarified. It likely arises from the dysregulated immune response of the host caused by the SARS-CoV-2 virus. It most often occurs in the postacute period of infection, and the clinical manifestation is heterogeneous, which makes diagnosis considerably more difficult in practice.

From a clinical point of view, it is essential to emphasize that unrecognized MIS-A has a high mortality rate, and that, for this very reason, it is necessary to start treatment immediately when the development of this disease is clinically suspected. In the early stage of the disease, the diagnosis of MIS-A is based exclusively on clinical symptoms and the patient’s history. Due to the danger of missing it, treatment should not be postponed to wait for the results of microbiological and serological examinations.

The prognosis of the disease depends on the early recognition of the condition and the rapid implementation of immunomodulating treatment (steroids, immunoglobulins, biological treatment), which reduces the risk of developing severe and life-threatening complications.

In the case of our patient, the development of MIS-A symptoms occurred two weeks after the diagnosis of COVID-19. The clinical picture was characterized by febrility, headache, maculopapular exanthema, myalgia, and arthralgia. The patient also had gastrointestinal symptoms such as diarrhea and abdominal pain for a short time; later, dyspnea and hypotension were added, which resembled a septic shock.

Damage to the cardiovascular system was mainly manifested by tachycardia, bigeminal ventricular extrasystoles on the ECG, as well as MRI-verified myocarditis with a decrease in the ejection fraction of the left ventricle and a laboratory elevation of TnT. After the administration of the immunomodulating treatment and the exclusion of focal infection, the clinical condition gradually improved, febrility receded, and laboratory parameters improved.

## Figures and Tables

**Figure 1 diagnostics-13-00983-f001:**
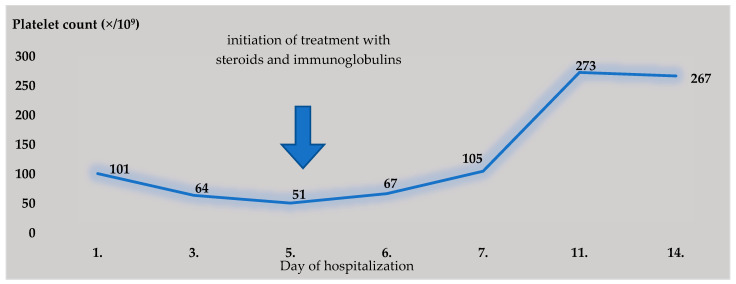
Level of platelets during hospitalization.

**Figure 2 diagnostics-13-00983-f002:**
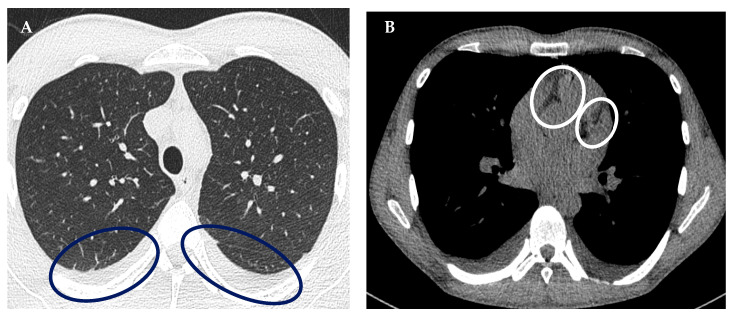
HRCT examination of the chest: (**A**)—Detection of fluidothorax bilaterally, up to 25 mm, more to the right (circle). (**B**)—Fluidopericarde up to 19 mm (circle).

**Figure 3 diagnostics-13-00983-f003:**
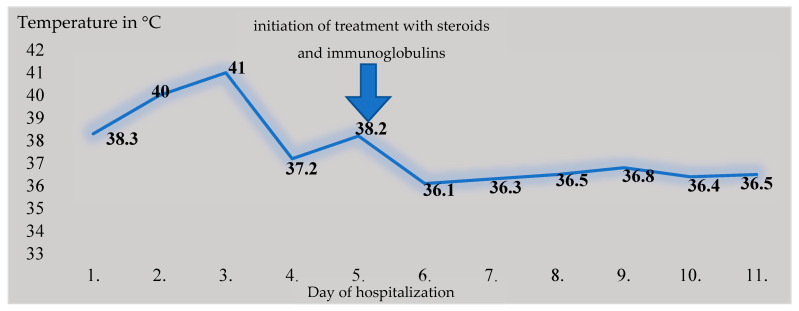
Temperature curve during hospitalization.

**Figure 4 diagnostics-13-00983-f004:**
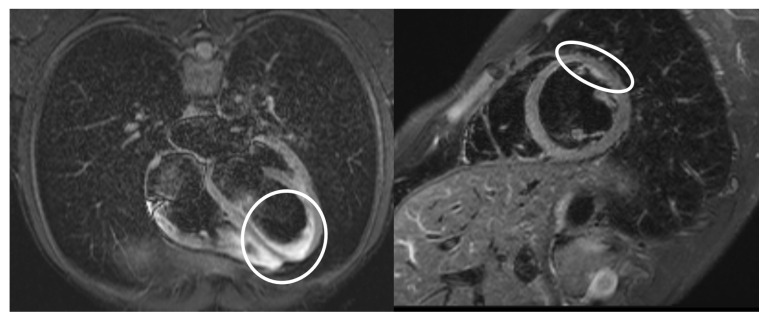
MRI of the heart—white circle showing the area of myocarditis.

**Table 1 diagnostics-13-00983-t001:** Results of performed hematological and biochemical examinations (*—an unexamined parameter on that day).

Day of Hospitalization	1.	3.	5.	6.	8.	13.	17.
Monitored Parameter							
C-reactive protein (CRP) (mg/L)	82.06	128.1	172.2	115.1	63.6	8.2	0.69
Procalcitonin (PCT) (ug/L)	0.24	0.73	1.19	0.73	0.34	0.05	0.04
Interleukin-6 (IL-6) (ng/L)	42.13	328.1	65.6	7.68	3.96	1.5	2.3
Lactate (mmol/L)	2.67	2.17	1.88	1.58	1.44	2.08	1.74
Creatinekinase (ukat/L)	1.2	1.5	0.8	0.52	0.41	0.15	1.2
Creatine kinase-MB (ukat/L)	0.17	0.14	0.16	0.19	0.20	0.29	0.22
Troponin T (ug/L)	0.003	0.007	0121	0.077	0.041	0.028	0.005
White blood cell (×10^9^/L)	6.36	3.26	4.80	7.21	9.55	7.26	8.79
Neutrophils (%)	82.6	82.1	91	91.7	93.2	89.2	77.9
Platelets (×10^9^/L)	101	64	51	67	105	273	267
D-dimer (mg/L)	0.43	1.5	1.48	2.26	2.40	1.96	0.46
NTproBNP (ng/L)	122	*	5438	8730	3709	*	125

**Table 2 diagnostics-13-00983-t002:** Results of performed microbiological examinations (*—repeated examination).

	Biological Material	Result
PCR SARS-CoV-2	Nasopharyngeal Swab	Pozit. Ct 32.2 Pozit. Ct 35.2 *
Anti-Epstein-Barr virus antibodies	serum	IgM negat./IgG pozit
Anti-Cytomegalovirus antibodies	serum	IgM negat./IgG negat. *
Anti-HIV virus antibodies	serum	negat.
Anti-*Chlamydia pneumoniae* antibodies	serum	IgM/IgA/IgG negat.
Anti-*Mycoplasma pneumoniae* antibodies	serum	IgM/IgA/IgG negat.
Hepatitis B surface *antigen*	serum	negat.
anti-Hepatitis C antibodies	serum	negat.
Anti-*Herpes simplex* 1,2 antibodies	serum	IgM negat./IgG pozit.
Anti-*Francisella tularensis* antibodies	serum	negat.
Anti-*Leptospira icterohemoragiae* antibodies	serum	negat.
*Candida/Aspergillus* antigen	serum	negat.
antigen *Legionella pneumophila/Streptococcus pneumoniae*	urine	negat.
PCR Influenza A/B	nasopharyngeal swab	negat.
adeno, rota, noroviruses antigen	stool	negat.
stool/rectal swab culture	stool	negat.
urine culture	urine	sterile *
*Clostridioides difficile*-toxin A/B, antigen	stool	negat.
blood culture	blood	sterile *

**Table 3 diagnostics-13-00983-t003:** Diagnostic algorithm for the definitive case of MIS-A (12).

Brighton Collaboration Case Definition	Patient from Our Case Report
**Age**	
Age <21 years (MIS-C) or ≥21 years (MIS-A)	22
**Fever**	
≥3 Consecutive days	6
**≥2 Clinical features**	
Mucocutaneous	Nonpurulent conjunctivitis, maculopapular exanthema on the chest
Gastrointestinal	Diarrhea, vomiting, and abdominal pain
Shock/hypotension	Hypotension, clinical signs of shock
Neurologic	Headaches
**Laboratory markers of inflammation**	
Elevated CRP	172.2 mg/L (maximal value)
Erythrocyte sedimentation rate	20 mm (after 1 h)/35 mm (after 2 h)
Elevated ferritin	556 ug/L (maximal value)
Elevated procalcitonin	1.19 ug/L (maximal value)
**≥2 Measures of disease activity**	
Elevated BNP or NTproBNP or Troponin T	NTproBNP: 8730 ng/L; Troponin T: 0.121 ug/L (maximal value)
Neutrophilia, lymfophenia, or thrombocytopenia	Neutrophils: 93.2%; Lymphocyte 3.30%; Platelets 51 × 10^9^/L
Echocardiographic evidence of cardiac involvement or physical sigmata of heart failure	Pericardial effusion
ECG changes consistent with myocarditis	Sinus tachycardia and bigeminal ventricular extrasystoles
**SARS-CoV-2**	
Laboratory confirmed SARS-CoV-2 infection or	Yes—PCR SARS-CoV-2 pozit. ct 32.2; ct 35.2
Personal historyof suspected COVID-19 within 12 weeks or	Yes
Close contact with known COVID-19 case within 12 week	Probably yes
OR	
SARS-CoV-2 vaccination	Yes—1 dose before 1 year

## Data Availability

Not applicable.

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
