# Peer review of "Multisystem Inflammatory Syndrome in Adults Associated with Recent Infection with COVID-19"

_diagnostics, 2023, doi:10.3390/diagnostics13050983_

Round 1

Reviewer 1 Report

MIS-A is an important issue, since it can be deadly, but if treated properly and timely patients should have complete recovery.

1. I suggest adding literature search and systematic review of other published cased so far to provide more comprehensive results 

2. I would suggest pro BNP values to be added or suggested to be performed in MIS-A patients

3. I would emphasize importance of the early treatment based solely on clinical grounds and patient history, without waiting for micrbiology and serology results

4. Emphasize possible of very high values of CRP and procalcitonin in MIS-A patients (personal experience with such patients)

5. Emphasize possible overlap in symptoms with toxic shock syndrome

Author Response

Dear reviewer, thank you for your professional comments and advice.

I incorporated all your comments into the new version of the manuscript.

1. We have added additional resources and a systemic review. 
2. We have added NTproBNP values to Table 1. 
3. In the text, we emphasized the importance of early treatment based on the patient's 
clinical picture and history without the need to wait for the results of microbiology 
and serology. 
4. We emphasized the presence of high inflammatory markers, which is characteristic 
of MIS-A. 
5. We emphasized possible overlap in symptoms with toxic shock syndrome. 

Best regards, authors

Reviewer 2 Report

MIS-A, as a complication of COVID-19, is rare and still of great clinical and public interest. Therefore, studies of individual clinical cases are relevant. In general, positively assessing the work, I have several comments and suggestions:

1. (115-116) Troponin T is not a marker of inflammation, but a marker of myocardial damage (no matter what nature), in this case insignificant (0.121 ug/l at N ≤ 0.1); NTproBNP is not a marker of inflammation, but a marker of heart failure (mainly left ventricular); markers of systemic inflammatory response are IL-6 and CRP, which were significantly elevated in the patient.

2. It should be indicated that the presence of foci of bacterial infection was excluded in the patient, since there are other criteria for sepsis (the presence of SIRS and probably signs of multiple organ dysfunction syndrome - MOD). To determine the degree of MOD, it was advisable to determine the values of the SOFA scale, especially since it was necessary during the use of intensive care.

3. It is not clear whether the presence of DIC syndrome was excluded in the presence of two positive criteria (D-dimer and platelets), it was also necessary to determine the level of fibrinogen and the value of prothrombin time (ISTH criteria).

4. There are signs of Epstein-Barr virus infection (presence of specific IgG); it is necessary to clarify whether this circumstance is an additional factor in the development of MIS-A, given that this virus has immunotropic activity. It is necessary to write Epstein-Barr Virus (EBV), not Ebstein-Barr virus (Table 2.).

5. Did the patient have signs of lymphadenopathy?

6. It is advisable to establish the level of diagnostic reliability of the clinical case in question according to the definition of the Brighton Collaboration for MIS-A and MIS-C [Vogel TP, Top KA, Karatzios C, Hilmers DC, Tapia LI, Moceri P, Giovannini-Chami L, Wood N, Chandler RE, Klein NP, Schlaudecker EP, Poli MC, Muscal E, Munoz FM. Multisystem inflammatory syndrome in children and adults (MIS-C/A): Case definition & guidelines for data collection, analysis, and presentation of immunization safety data. Vaccine. 2021 May 21;39(22):3037-3049. doi: 10.1016/j.vaccine.2021.01.054. Epub 2021 Feb 25. PMID: 33640145; PMCID: PMC7904456].

Author Response

Dear reviewer, thank you for your professional comments and advice.

I incorporated all your comments into the new version of the manuscript.

The patient from our case report did not have lymphadenopathy.

You are correct that the EBV virus is immunotropic; however, our patient only had isolated positive IgG antibodies against this virus, which means that he overcame the infection in the distant past, which means that the EBV virus did not affect the development of MIS-A.

  1. We corrected mistakes, TnT and NTproBNP are not markers of inflammation.
  2. In the corrected manuscript, we added that the microbiological and imaging examinations performed did not confirm a focal bacterial infection, but even so, its presence could not be ruled out with certainty, so we continued the treatment with broad-spectrum antibiotics.
  3. We added to the manuscript that DIC was excluded based on ISTH criteria. Fibrinogen and prothrombin time levels were normal (ISTH score 2).
  4. The patient from our case report did not have lymphadenopathy.
  5. You are correct that the EBV virus is immunotropic; however, our patient only had isolated positive IgG antibodies against this virus, which means that he overcame the infection in the distant past, which means that the EBV virus did not affect the development of MIS-A.
  6. We determined the diagnostic confidence level of the respective clinical case according to the Brighton Collaboration definition for MIS-A and MIS-C.

Best regards, authors

Reviewer 3 Report

I read with interest this case report concerning a possible case of multisystem inflammatory syndrome in adults associated with 2 recent infection with COVID-19. The topic is quite interesting, but I have several major comments:

1.       Lines 49-53. “The most important clinical symptom is a 49 fever above 38°C lasting more than 24 hours before hospitalization or during the first three 50 days, and at least three of the clinical criteria occur within 24 hours. Before or during the 51 first three days of hospitalization, at least one of them must be the primary criterion (tab. 52 1) (4).” Comment: This definition does not seem to be clear. Please consider rephrasing.   

2.       Lines 112-114 and lines 131-133: The lung ultrasonography revealed B-lines, yet the HRCT did not show “interstitial lung edema, and without cardiac undercompensation in the event of a probable myopericarditis”. How do the authors explain this discrepancy.

3.       Line 164: How was the addition of colchicine to corticosteroid treatment justified?

General comment: The patient received broad-spectrum antibiotics from the beginning, and this treatment was later escalated before improvement was noticed. How can the authors exclude the possibility of a bacterial or other infection perhaps super-imposed to the previous SARS-COV-2 infection? I think this deserves further discussion.

Author Response

Dear reviewer, thank you for your professional comments and advice.

I incorporated all your comments into the new version of the manuscript.

The difference between the finding on lung ultrasound and the finding on HCRT is mainly explained by the time gap between the given imaging methods. We performed USG of the lungs at a time when the patient had dyspnea, and B lines were a sign of interstitial edema. At that time, fluidothorax was not present. We did HRCT already after starting the immunomodulating treatment (4 days later than the USG of the lungs).

  1. We reformulated the definition of MIS-A.
  2. The difference between the finding on lung ultrasound and the finding on HCRT is mainly explained by the time gap between the given imaging methods. We performed USG of the lungs at a time when the patient had dyspnea, and B lines were a sign of interstitial edema. At that time, fluidothorax was not present. We did HRCT already after starting the immunomodulating treatment (4 days later than the USG of the lungs).
  3. Colchicine was added to the treatment on the recommendation of an internist as an additional treatment for myopericarditis.

In the corrected manuscript, we added that the microbiological and imaging examinations performed did not confirm a focal bacterial infection, but even so, its presence could not be ruled out with certainty, so we continued the treatment with broad-spectrum antibiotics.

Best regards, authors

Round 2

Reviewer 1 Report

no suggestions

Reviewer 3 Report

All my comments have been addressed satisfactory.